# Personalized Federated Learning Algorithm with Adaptive Clustering for Non-IID IoT Data Incorporating Multi-Task Learning and Neural Network Model Characteristics

**DOI:** 10.3390/s23229016

**Published:** 2023-11-07

**Authors:** Hua-Yang Hsu, Kay Hooi Keoy, Jun-Ru Chen, Han-Chieh Chao, Chin-Feng Lai

**Affiliations:** 1Shenzhen Graduate School, Peking University, Beijing 100191, China; huayang.hsu@gmail.com; 2UCSI Graduate Business School, UCSI University, Kuala Lumpur 56000, Malaysia; keoykh@ucsiuniversity.edu.my; 3Department of Engineering Science, National Cheng Kung University, Tainan 70101, Taiwan; ccb1147l@gmail.com; 4Department of Electrical Engineering, National Dong Hwa University, Hualien 974301, Taiwan

**Keywords:** Non-IID IoT data, adaptive clustering, personalized federated learning

## Abstract

The proliferation of IoT devices has led to an unprecedented integration of machine learning techniques, raising concerns about data privacy. To address these concerns, federated learning has been introduced. However, practical implementations face challenges, including communication costs, data and device heterogeneity, and privacy security. This paper proposes an innovative approach within the context of federated learning, introducing a personalized joint learning algorithm for Non-IID IoT data. This algorithm incorporates multi-task learning principles and leverages neural network model characteristics. To overcome data heterogeneity, we present a novel clustering algorithm designed specifically for federated learning. Unlike conventional methods that require a predetermined number of clusters, our approach utilizes automatic clustering, eliminating the need for fixed cluster specifications. Extensive experimentation demonstrates the exceptional performance of the proposed algorithm, particularly in scenarios with specific client distributions. By significantly improving the accuracy of trained models, our approach not only addresses data heterogeneity but also strengthens privacy preservation in federated learning. In conclusion, we offer a robust solution to the practical challenges of federated learning in IoT environments. By combining personalized joint learning, automatic clustering, and neural network model characteristics, we facilitate more effective and privacy-conscious machine learning in Non-IID IoT data settings.

## 1. Introduction

In recent years, federated learning has emerged as a prominent research domain at the crossroads of artificial intelligence and decentralized data processing. The ever-evolving landscape of AI and machine learning necessitates a critical reassessment of the state-of-the-art in federated learning to ascertain its significance and potential applications [1]. However, the effective implementation of federated learning often hinges on the availability of extensive and diverse datasets, traditionally assumed to be Independent and Identically Distributed (IID). Challenges surface when dealing with data that deviate from this IID assumption, a common occurrence in scenarios involving Internet of Things (IoT) devices [2]. Traditional federated learning approaches encounter difficulties in harnessing data from diverse IoT devices, each characterized by its unique data distribution and features. In the context of federated learning, it is imperative to adapt and evolve to effectively handle the increasingly complex, dynamic data sources, and decentralized IoT systems [3].

Concerning the issue of complex, dynamic data sources, hybridizations of distributed policy and heuristic augmentation represent pivotal components that can augment the federated learning approach by amalgamating the strengths of policy-based strategies with heuristic-based techniques [4]. Policy-based approaches excel at learning flexible and adaptive strategies but may necessitate substantial data to attain robustness. Conversely, heuristic-based techniques offer stability and guidance in scenarios with limited data, thereby mitigating the risks associated with data scarcity and heterogeneity [5]. The fusion of these two approaches yields a versatile framework capable of adapting to varying data conditions and optimizing model performance [6]. On the other hand, integrating fuzzy consensus within federated learning methods presents a complex challenge when striving to achieve consensus in a decentralized environment, particularly in the presence of Non-IID data sources [7]. Fuzzy consensus introduces the concept of graded agreement, acknowledging that complete uniformity is often unattainable in real-world data scenarios. This approach considers the uncertainty and variability inherent in data collected from IoT devices, accommodating these aspects through a more flexible consensus mechanism [8]. Embracing fuzzy consensus equips federated learning to better handle the inherent noise and diversity in Non-IID IoT data. It enables the system to reach agreements adaptable to the varying characteristics of data sources, thereby enhancing the resilience and reliability of the federated learning process in the face of uncertainty [9,10]. However, the realization of this ideal design has faced several practical challenges. Notably, four key challenges have arisen in the domain of federated learning:Communication Cost: As federated learning inherently involves distributed learning, the transmission of training data, such as model parameters, to participating clients remains a substantial cost factor [11]. Even though federated learning does not necessitate the exchange of raw data between devices, the involvement of a large number of devices in the training process still incurs significant communication overhead [12]. Addressing this issue involves optimizing data transmission to maintain model accuracy while minimizing data size, reducing transmission frequency, and enhancing communication efficiency.Equipment Heterogeneity: The performance capabilities of devices participating in federated learning can vary significantly, leading to issues like stragglers or dropout effects. Stragglers, devices with limited storage, computational power, or communication capabilities, can substantially prolong the duration of a federated learning round [13]. While discarding these devices may result in information loss, their asynchronous communication can disrupt the current training model. Dealing with equipment heterogeneity requires high fault tolerance or carefully selecting compatible devices for training.Privacy Issues: Federated learning aims to address privacy concerns inherent in traditional distributed learning. It achieves heightened privacy by transmitting only the data necessary for training, avoiding the exchange of personally sensitive information [14]. However, even though private data are not directly transmitted, adversaries can potentially infer training data from the information transmitted back to the server, such as model parameters and gradients. Balancing privacy protection with model accuracy and system efficiency becomes a critical consideration.Non-IID Data: Federated learning faces the challenge of non-independent and non-identically distributed (Non-IID) data, in contrast to traditional machine learning’s assumptions of IID data [15,16]. In federated learning, each device’s data are sampled from different distributions and may exhibit dependencies. Non-IID data directly impact the final model’s accuracy, complicate the training process, and impede convergence. Addressing this challenge is pivotal for advancing federated learning.

In this paper, we introduce an innovative clustering approach designed to address the challenges associated with Non-IID data within the federated learning framework. By grouping clients with similar data distributions, our approach aims to enhance the efficiency of federated learning without imposing additional burdens on the existing methodology. Our method significantly improves the accuracy of the final model in Non-IID scenarios, as demonstrated through comparative analyses with FedAvg and other clustering methods across various Non-IID settings. Our contributions in this paper encompass:Clustering Approach: We propose a clustering approach integrated into the federated learning framework to address the Non-IID data challenge.Experimental Design: We establish multiple Non-IID environments for comparative evaluations, offering insights into the strengths and weaknesses of our approach.Future Directions: We identify potential areas for further improvement in our method, facilitating future research in this domain.

In summary, our research endeavors aim to enhance the applicability and effectiveness of federated learning in addressing contemporary challenges. We aspire to advance AI development while safeguarding privacy and data integrity.

## 2. Related Works

### 2.1. Personalized Federated Learning

In the rapidly evolving landscape of distributed learning, numerous research efforts have made substantial contributions to the optimization of algorithms and the enhancement of communication efficiency. These studies have addressed a wide range of challenges associated with scaling machine learning algorithms to distributed environments, ensuring convergence guarantees, and enabling effective communication. Notable among these contributions is the work of Boyd et al., which delved into the realm of distributed convex optimization, pushing its boundaries to extend applicability into non-convex settings, thereby unlocking new possibilities for versatile optimization techniques [17].

In a parallel vein, Dekel et al. introduced a distributed mini-batch algorithm designed to expedite the training speed of distributed learning models, resulting in significant linear acceleration benefits [18]. In the pursuit of scalability for large-scale distributed learning, Dean et al. devised two innovative algorithms that addressed the intricacies of managing extensive datasets and distributed computing environments [19]. Zhang et al. complemented these efforts by proposing two efficient communication algorithms explicitly tailored for large-scale distributed learning applications, further enhancing scalability and efficiency [20]. Li et al. embarked on a crucial journey, expanding the scope of distributed learning to encompass non-convex problem domains [21]. Their work led to the development of a third-generation parameter server framework that provides convergence guarantees even in non-convex scenarios, offering robust solutions for a broader range of optimization challenges. Furthermore, Shamir et al. tackled the issue of slow convergence in distributed learning by developing a method capable of accelerating linear convergence rates based on data size, significantly improving the efficiency of distributed learning [22]. Additionally, Zhang et al. introduced the concept of EASGD, a flexible approach that facilitates seamless coordination among local devices in distributed learning [23]. EASGD supports both synchronous and asynchronous settings, providing adaptability to diverse distributed learning scenarios. Reddi et al. made a significant contribution by introducing AIDE, a framework that seamlessly combines fast convergence capabilities with efficient communication algorithms [24]. This novel approach addresses the dual challenges of speed and communication efficiency in distributed learning. Richtárik and Takáč explored clustering tasks based on data features, expanding the applicability of distributed learning to diverse data analysis domains through the introduction of Hydra, a distributed algorithm focused on clustering [25]. Lastly, Smith et al. presented CoCoA, an efficient communication scheme designed to handle non-convex problems effectively, thereby offering robust optimization capabilities in distributed learning scenarios [26]. The following are key issues that are often raised:Communication Cost Reduction:

Efficiently reducing communication costs represents a critical aspect of optimizing federated learning systems. Researchers have explored two primary directions to achieve this: data compression and minimizing the frequency of communication.

2.Data Compression:

An effective approach for reducing communication overhead involves data compression techniques. Model compression has proven to be particularly efficient in this regard. One practical method is to operate within a low-dimensional space by applying techniques such as low-rank approximation or random masking to model gradients [27]. Alternatively, model gradients can be compressed using methods such as subsampling, probabilistic quantization, or structured random rotations, all of which are lossy compression techniques. While these methods can significantly decrease communication costs, it is important to note that as the compression rate increases, model accuracy tends to decrease. However, many existing compression methods are sensitive to Non-IID data, making them less suitable for federated learning environments. To address this, sparse ternary compression was specifically designed to handle Non-IID data. Additionally, the adaptive algorithm ACFL adjusts compression rates based on communication conditions, addressing issues related to network reliability and dynamic bandwidth in federated learning [28]. It should be noted that the effectiveness of ACFL might depend on the underlying system architecture.

3.Communication Methods:

In federated learning, communication methods play a vital role in determining system efficiency. There are two main communication methods: synchronous and asynchronous. In synchronous communication, all participating devices must transmit their data before the aggregation process can begin. While this approach ensures data consistency, it is vulnerable to the impact of stragglers, significantly prolonging training time. One commonly used algorithm in this context is Federated Averaging (FedAvg) [29]. FedAvg speeds up model training by conducting multiple rounds of local computation before communication, effectively reducing the number of communication rounds. However, it is worth noting that FedAvg lacks a convergence guarantee for Non-IID data. Asynchronous communication, on the other hand, does not focus on reducing the number of communications but is better suited to handle equipment heterogeneity. While asynchronous communication may not be as effective in minimizing communication frequency, it offers advantages in scenarios where devices have varying capabilities. To further enhance communication efficiency, researchers have explored advanced algorithms like test-based FedAvg. This approach evaluates uploaded gradient parameters and transmits only those updates that meet predefined evaluation criteria, thus reducing communication volume and mitigating the impact of Non-IID data on training. However, determining the appropriate criteria for evaluating the usefulness of return parameters remains an open question in this context.

### 2.2. Data Heterogeneity for Federated Learning

In the field of federated learning, researchers have undertaken extensive efforts to tackle the challenges posed by data heterogeneity, particularly in scenarios involving non-identically and independently distributed (Non-IID) data. These endeavors encompass global and personalized strategies, each tailored to mitigate the impact of data heterogeneity and enhance the efficiency and efficacy of federated learning algorithms.

Global Strategies:

Global strategies are designed to ensure model consistency across all participating clients, thereby preventing divergence and promoting convergence. FedProx introduces a proximal term in local client updates to maintain model parameters in proximity to the global model. While it succeeds in preserving model consistency, it may exhibit slower convergence rates in cases where the data are independent and identically distributed (IID). In response to weight divergence issues, SCAFFOLD incorporates control variates in local client updates to adjust gradient deviations from the global gradient. However, this method comes at the cost of increased communication overhead and client resource utilization. Qu demonstrated that self-attention-based architectures (e.g., Transformers) enhance federated learning over heterogeneous data, addressing challenges like convergence and catastrophic forgetting, through a rigorous empirical investigation across various federated algorithms, real-world benchmarks, and heterogeneous data splits. [30]. In response to the challenge of data heterogeneity in personalized federated learning, Dai proposed FedNH, a novel method that enhances both personalization and generalization of local models by improving class prototype uniformity and semantics, ultimately showing its effectiveness and stability across various classification datasets [31]. Weight assignment based on these contributions expedites neural network model convergence, reducing the overall number of required communication rounds. Nonetheless, this approach may introduce biases towards data distribution with higher contributions. Addressing parameter variation across different client data distributions, Mendieta et al. a simple and efficient method that competes with state-of-the-art FL approaches while minimizing computation and memory overhead [32].

2.Personalization Strategies:

Personalization strategies acknowledge the diversity of client data and aim to create individualized models for each client, ultimately aligning the models more closely with the respective data distributions. Deng et al. advance TailorFL that is a dual-personalized federated learning framework that tailors sub-models to individual devices based on their resource capabilities and local data distribution, resulting in improved inference accuracy and reduced resource usage compared to existing methods [33]. Tan delved into the realm of personalized Federated Learning (PFL) to tackle the challenges of training machine learning models on heterogeneous data while preserving privacy, offering insights into key motivations, techniques, challenges, and envisioning future research directions in the field [34]. This minimizes communication costs and enables distinct local models to learn the same output. Yang et al. introduce pFedPG, a novel personalized Federated Learning (FL) framework that leverages large-scale pre-trained models while efficiently personalizing models for heterogeneous clients by deploying client-specific prompt generators [35]. They employ fine-tuning with local data to enhance the performance of personalized models. Qin et al. introduce FedAPEN, a novel Federated Learning (FL) framework that combines mutual and ensemble learning, allowing for heterogeneous models and improving accuracy on diverse data distributions, even in scenarios where data heterogeneity is immeasurable due to privacy concerns [36]. During aggregation, only the base layer is sent to the server, reducing communication overhead while facilitating personalization. Li et al. introduce FedTP, a novel transformer-based federated learning framework that addresses the negative impact of federated averaging (FedAvg) on self-attention in cases of data heterogeneity, thus enhancing the capabilities of transformer models in federated learning settings [37]. Shen et al. propose a novel framework for personalized federated learning that addresses data heterogeneity challenges by implementing channel-wise personalization, known as channel decoupling, and employing a cyclic distillation scheme to promote consistency between local and global model representations [38]. Cao et al. introduce a federated learning method based on co-training and generative adversarial networks (GANs), allowing individual clients to independently design their own models for federated learning without sharing architecture or parameter information [39]. They also provide generalization boundaries for these approaches. Wu et al. design a cloud-edge architecture for personalized federated learning [40]. This architecture optimizes computation speed via computation offloading and minimizes communication costs through edge-to-cloud server transfers.

These research efforts collectively contribute to the advancement of federated learning, particularly in scenarios involving data heterogeneity. By offering a spectrum of strategies, both global and personalized, these approaches help address challenges related to data distribution diversity, ultimately enhancing the efficiency and effectiveness of federated learning algorithms.

## 3. Personalized Federated Learning Algorithm

Federated learning is an innovative approach that enables collaborative model training across multiple client devices while ensuring data privacy. The architecture of federated learning comprises two main components: the server-side and the client-side. On the server-side, there exists a central server that manages the global neural network model’s parameters. Importantly, the server does not have direct access to the training data. The primary responsibilities of the server include selecting clients to participate in federated learning rounds, transmitting model parameters and necessary hyperparameters to the selected clients, and aggregating and updating the global model based on the contributions from the clients. On the client-side, each participating client possesses its own training data. Clients perform two key tasks. First, they wait for the server to send the parameters of the neural network model, and during this waiting period, clients collect and store data locally. Second, after receiving the model parameters from the server, clients use their local data to train the model. Once the training is completed, clients send back the trained model parameters or gradients to the server. A typical federated learning round consists of four steps:Selection and Broadcast:

The server selects clients to participate in the federated learning round. It encrypts and transmits the model parameters and other necessary training-related parameters to the selected clients.

2.Local Training:

Clients receive the model parameters and perform local gradient descent on the neural network model using their respective local data.

3.Upload Parameters:

After local training, clients send the trained model parameters back to the server. This step also involves encryption for privacy.

4.Aggregation and Update:

The server aggregates the model parameters or gradients received from the clients based on a predefined aggregation strategy. Subsequently, it updates the global neural network model with the aggregated parameters.

These four steps are repeated multiple times throughout the federated learning process, either until the model converges or for a specified number of training sessions.

### 3.1. Formula

Federated learning aims to minimize a global objective function that aggregates local objective functions across clients. The mathematical formulation of federated learning can be described as follows:(1)FωMin(ω)=∑i=1NpkFk(ω)

The global objective function, denoted as F(ω) represents the goal of federated learning, where ω represents the parameters of the neural network model. In this formula, F(ω) is expressed as the sum of local objective functions $F_{k}\left(\omega \right)$ weighted by coefficients $p_{k}$.
(2)Fk(ω)=ℒωargminxi,yi~Dk(xi,yi;ω)

These coefficients, $p_{k}$ reflect the contribution of each client to the global objective. Typically, $p_{k}$ is determined as pk=nkn where $n_{k}$ represents the size of the dataset at the kth client, and n is the total number of data points across all clients. Local objective functions, $F_{k}\left(\omega \right)$ are similar to those used in traditional machine learning and are applied to measure the divergence in data distribution for each client. There are $x_{i}$ and $y_{i}$ which sample from the input space Χ and the output space Υ, respectively. Thus, *x* ∈ *X* and *y* ∈ *Y*. We combine them to be a pair of labeled data (*x*, *y*) ∈ *X ×*
Υ. We denote the sampling space of *X ×*
Υ by a distribution *D*. $D_{k}$ is a distribution for the *k*th client. ℒ is loss function used at the client-side. In FedAvg, the loss function will be the same for each client.
(3)$$\omega_{k,\tau +1}^{t}=\omega_{k,\tau }^{t}-\eta \nabla F_{k}\left(\omega_{k,\tau }^{t}\right)$$

Commonly, loss functions such as cross-entropy loss are utilized for classification problems, t is the federated learning round, and stochastic gradient descent techniques are employed for optimization. To enhance federated learning efficiency, the FedAvg algorithm introduces the concept of multiple local epochs E on the client-side. Each federated learning round consists of several local iterations. Additionally, parameters like the learning rate η influence the training process and τ is the current epoch in the local training. This mathematical framework forms the basis of federated learning, facilitating collaborative model training while accommodating diverse data distributions among clients and ensuring data privacy. At the beginning of each federated learning round, the local model parameter $\omega_{k}$ will be initialized on the basis of the global model parameter ω sent from the server, if the federated learning round is successful at last time. In one federated learning round, Equation (3) can be simplified to Equation (4) without epoch.
(4)$$\omega_{k}^{t+1}=\omega^{t}-\eta \nabla F_{k}\left(\omega^{t}\right)$$

When the client finishes local training, it sends the result of the training to the server. The server aggregates all of the result returned by clients using aggregate weight.
(5)$$\omega^{t+1}=\sum_{k=1}^{N}\frac{n_{k}}{n}\omega_{k}^{t+1}$$

### 3.2. Algorithm

In the previous section, we discussed the limitations of existing federated learning methods, particularly in handling Non-IID (Non-Independently and Identically Distributed) data. These methods primarily rely on weight aggregation to align local and global objectives for accelerated convergence. However, when data distributions on client devices are highly skewed, these skewed gradient updates receive very low weights in the aggregation process. This inefficiency can lead to wasted computational resources and hinder the federated learning system’s overall efficiency.

To address this challenge, we introduce a novel approach, federated cluster, which utilizes clustering to group clients with similar data distributions. The goal is to achieve accelerated convergence while ensuring efficient resource utilization. Before delving into the formal introduction of federated clusters, we provide an overview of the FedAvg algorithm to highlight the differences and showcase how our approach can enhance model accuracy with minimal additional complexity.

Algorithm Federated Averaging (FedAvg) describes the pseudocode of FedAvg in Algorithm 1, a commonly used federated learning algorithm. It involves initializing the neural network model and selecting a group of clients for training in each round. The global model parameters are transmitted to the selected clients, who then perform local gradient updates. After training, clients send back their updated model parameters, and the server aggregates these parameters based on the amount of data owned by each client.
**Algorithm 1** Federated Averaging (FedAvg)**Server-side****Input:***N*, *T*, *C*
  1: Initialize ω0

  2: **for** each round t = 0, 1, ...T **do**

  3:       m ← max(N·K, 1)

  4:       St  ← (sample m clients from C)

  5:       **for** each client k ∈ St **do in parallel**

  6:         ωkt+1 ← **ClientUpdate**(ωt)

  7:       $\omega^{t+1}$ ← $\underset{k\;\in \;S^{t}}{\sum }p_{k}\omega_{k}^{t+1}$

  8: **return**
$\omega^{t+1}$

**Client-side**

**ClientUpdate**($\omega^{t}$)

**Input:**
*E, η*

  1: **for**
*τ* in epoch *E*
**do**

  2:   ωk,τt *←*ωk,τ−1t− η∇Fk(ωk,τ−1t)

  3: **return**
$\omega_{k,\tau }^{t}$


While FedAvg serves as the foundation for federated learning, it has a fundamental limitation when dealing with skewed data distributions. If a client possesses significantly more data than others but follows a Non-IID distribution, it can bias the global model towards its skewed distribution, hindering performance on other clients’ data. This is a critical drawback of the weight aggregation strategy, which may lead to suboptimal model generalization. In light of the challenges posed by Non-IID IoT data, we introduce a novel clustering algorithm called Federated Adaptive Cluster (FedAC) as part of our proposed personalized federated learning approach in Algorithm 2. The primary objective of FedAC is to address the issue of clients with smaller contributions to Federated Learning. These clients, which may have highly skewed or less representative data, can hinder the overall efficiency of the federated learning system.
**Algorithm 2** Federated Adaptive Cluster (FedAC)**Server-side****Input:***N*, *T*,*C*,*α*,*β*
  1: Initialize ω0

  2: **for** each round t = 0, 1, ...T **do**

  3:       m ← max(N·K, 1)

  4:       St  ← (sample m clients from C)

  5:       **for** each client k ∈ St **do in parallel**

  6:         ωkt+1 ← **ClientUpdate**(ωt)

  7:       ωt+1 ← ∑k ∈ Stpkωkt+1

  8:       Δloss ← evaluate($\omega^{t+1}$) − evaluate($\omega^{t}$)

  9:       θ ← $\min\left(\cos\mathrm{ine}\_\mathrm{similarity}\left(\Delta \omega^{t+1},\;\Delta \omega_{k}^{t+1}\right)\right)$, k ∈ $S^{t}$

  10:       if |Δloss| < α and θ < β **then**

  11:         $C_{1}$, $C_{2}$ ← Cluster (St, C)

  12 **return** $\omega_{k}^{t+1}$

**Cluster** (St, C)

  1: ∆$w_{\mathrm{right}}$, ∆$\omega_{left}$ ← find_max(cosine_similarity($\Delta \omega_{i}^{t+1}$, $\Delta \omega_{j}^{t+1}$)), k ∈ $S^{t}$

  2: **for** each client k ∈ C **do**

  3:   **if** cosine_similarity($\Delta \omega_{k}^{t+1}$, ∆$w_{\mathrm{right}}$) > cosine_similarity($\Delta \omega_{k}^{t+1}$, ∆$w_{\mathrm{left}}$)

**then**

  4:       assign client k →$C_{1}$

  5:   **else**

  6:       assign client k → $C_{2}$

  7: **return** $C_{1}$, $C_{2}$

**Client-side**

**ClientUpdate**($\omega^{t}$)

**Input:**
*E*, *η*

  1: **for**
*τ* in epoch *E*
**do**

  2:   ωk,τt *←*ωk,τ−1t− η∇Fk(ωk,τ−1t)

  3: **return**
$\omega_{k,\tau }^{t}$


Our clustering approach allows us to harness their potential effectively. The key aspect of our clustering algorithm is the strategy for separating clients. We acknowledge that the initial model parameters and distance threshold settings can significantly influence the accuracy of client clustering. To mitigate this, we have opted to delay the clustering process until after a certain amount of training has been completed. This strategic decision ensures that the model parameters have reached a suitable state for accurate clustering, as illustrated in Figure 1. We outline the key steps and components of the FedAC algorithm:

Input Parameters: FedAC introduces additional parameters compared to FedAvg. The input parameters include N (proportion of clients participating), T (total training rounds), and C (the set of clients in a cluster).Client Selection: Clients to be trained in each round are sampled from the cluster based on the defined proportion of participating clients (N).Local Training: Selected clients receive the global model from the server and perform local updates using gradient descent. This step ensures that the algorithm does not increase the computational burden on client devices, aligning with the federated learning principle of minimizing client resource consumption.Aggregation with Evaluation: After clients complete local training and send back their models, the server aggregates these models. Importantly, FedAC introduces an evaluation step. The server assesses whether the current global model is close to convergence or has already converged.Clustering Decision: The decision to perform clustering relies on two conditions. Firstly, we assess the degree of loss reduction to determine the proximity of the global model to convergence. Secondly, we employ cosine similarity to evaluate the correlation between client gradient updates and the gradient update of the global model. Cosine similarity measures the angle and direction of these updates, helping us identify clients whose gradient updates significantly deviate from the direction of the global model. Such deviations may indicate variations in data distributions among clients.Clustering: If clustering is warranted, the server selects a subset of clients for clustering based on their gradient updates. The selected clients are divided into two clusters using a dichotomy approach. This process is designed to efficiently group clients with distinct data distributions.

The clustering process shown in Figure 2 is not dependent on setting specific distance thresholds or using the global model as a basis for classification. Instead, it selects clients with the farthest angular distance from each other and proceeds to divide clients into clusters iteratively. The FedAC aims to cluster clients intelligently after substantial training, ensuring that the model parameters are in a position to be accurately grouped. By doing so, it mitigates issues related to skewed data distributions, accelerates convergence, and maximizes resource utilization.

## 4. Experiment Results

We delve into the experimental results, presenting our findings based on various parameter settings, datasets, and model structures. We also provide insights into the effectiveness of our proposed algorithm.

### 4.1. Dataset

We utilized the CIFAR-10 dataset for training and classification tasks in our federated learning experiments [41]. CIFAR-10 comprises color images and boasts the following specifications:Number of Classes: 10;Total Dataset Size: 60,000 images (6000 images per class);Training Dataset Size: 50,000 images;Test Dataset Size: 10,000 images.

Despite CIFAR-10′s inherently IID (Independently and Identically Distributed) nature, we introduced Non-IID scenarios for federated learning. Our Non-IID scenarios aimed to simulate more realistic data distributions across clients, considering factors such as geographical location, age, and social environment. We categorized the data distributions into four distinct cases:IID Situation: Clients have the same probability of having data from all classes.Non-IID Situation (two clusters): Clients are divided into two clusters, each with five classes of data.Non-IID Situation (three clusters): Clients are divided into three clusters, with two clusters containing three classes of data and one cluster with four classes of data.Non-IID Situation (four clusters): Clients are divided into four clusters, with two clusters containing two classes of data and two clusters containing three classes of data.

This classification allowed us to examine how our algorithm performs across varying degrees of data distribution heterogeneity among clients.

### 4.2. Experimental Design

We outline the hyperparameters used in our experiments, implemented using the Flower framework. Key hyperparameters include:Number of Clients: 100;Total Number of Rounds: 200;Proportion of Participating Clients: 0.1;Local Epochs: 5;Learning Rate: 0.001;Momentum: 0.9;Batch Size: 32;Clustering Thresholds (α and β): α is set to 5 for assessing convergence, while β employs a 90-degree threshold for detecting inconsistent data distributions within clusters.

### 4.3. Experimental Results

#### 4.3.1. IID Situation

In the IID scenario, we observe that the performance of FedAC, FedACP, and FedAvg is nearly identical in Figure 3, which allows us to assess the time complexity of each method. Both accuracy and loss closely overlap in their performance curves, indicating that these methods exhibit similar time complexities. This similarity can be attributed to the fact that FedAC utilizes the same aggregation and training methods as FedAvg, with the primary difference being the clustering action. In the IID case, FedAC does not perform redundant clustering, resulting in performance equivalent to FedAvg.

#### 4.3.2. Non-IID Situation

In the Non-IID scenarios, we observe distinct differences in performance. FedAvg exhibits significant oscillations in both accuracy and loss due to the inherent Non-IID nature of the data distributions among clients. In contrast, FedAC stabilizes quickly after clustering, as evidenced by Figure 4 and Figure 5. We can observe that clustering in FedAC and FedACP impacts the time and computational complexity of the algorithms. Figure 6 and Figure 7 show that as the number of client clusters increases, the frequency of clustering in FedAC also rises, which translates to higher computational resource consumption. Reducing resource consumption during clustering will be a crucial consideration in future enhancements, and it affects the computational complexity of the algorithms. FedACP, while not consistently accurate in clustering, offers a potential alternative for reducing communication costs during clustering while providing performance improvements in model accuracy. These findings not only demonstrate the effectiveness of our proposed Federated Cluster algorithm in addressing the challenges posed by Non-IID data distributions among clients but also shed light on the time and computational complexities associated with these methods. Further quantitative analysis will be provided in the revised version of the paper to provide a comprehensive view of the time and computational aspects of our approach.

Figure 6 and Figure 7 indicate that as the number of client clusters increases, the frequency of clustering in FedAC also rises, which translates to higher computational resource consumption. Reducing resource consumption during clustering will be a crucial consideration in future enhancements. FedACP, while not consistently accurate in clustering, offers a potential alternative for reducing communication costs during clustering while providing performance improvements in model accuracy. The oscillations in performance following the last clustering eventually subside. These findings demonstrate the effectiveness of our proposed Federated Cluster algorithm, particularly in addressing the challenges posed by Non-IID data distributions among clients in the federated learning context.

## 5. Conclusions

Our algorithm demonstrates robustness and effectiveness in transforming Non-IID data environments into IID-like scenarios, making it particularly valuable in privacy-sensitive applications. In healthcare scenarios, for instance, where patient habits may significantly impact the data distribution, our FedC algorithm allows us to create separate training groups without revealing the sensitive habits of individual patients. However, while discussing the capabilities of our algorithm, it is important to highlight the inherent privacy and security aspects of Federated Learning, which have been an integral part of our work. We outline key areas for future research and development to further enhance the capabilities and robustness of our algorithm. Our current clustering initiation method relies on straightforward threshold-based criteria. Future research should focus on devising more sophisticated and adaptable detection methods. These methods should be capable of accurately identifying the optimal timing for clustering in scenarios with oscillating loss variations, thus improving the clustering process’s efficiency. While clustering generally leads to improved model performance, there are instances where clustering does not yield optimal results, resulting in suboptimal or inefficient clusters. Future work should concentrate on enhancing the efficiency and effectiveness of the clustering process. This may involve refining the clustering algorithms or introducing adaptive mechanisms to adjust to changing data distributions. Scanning all clients within a cluster during each clustering round can be computationally demanding. Future efforts should aim to optimize the computational efficiency of the clustering process. This includes reducing computation time and addressing potential issues caused by straggling clients. While our algorithm currently utilizes Federated Averaging (FedAvg) for aggregation and training, it is worthwhile to explore alternative global strategies such as Federated Proximal (FedProx). Investigating these alternative strategies can help determine if they offer performance improvements in specific scenarios. Conducting additional experiments is essential to evaluate the benefits of alternative global strategies. The use of model parameters for clustering may raise privacy concerns, particularly in scenarios involving sensitive data. Future research should focus on developing techniques that ensure effective clustering while preserving privacy. Differential privacy and other privacy-preserving methods should be explored to mitigate privacy risks associated with clustering. Lossy compression techniques used for communication may impact efficiency. Future work should investigate approaches to minimize the communication overhead caused by lossy compression. This involves optimizing compression algorithms and strategies to improve communication efficiency without compromising data integrity. By addressing these recommendations, researchers and practitioners can further refine and expand the capabilities of our personalized federated learning algorithm with adaptive clustering. This will make the algorithm more adaptable and robust, particularly in the context of Non-IID IoT data scenarios.

## Figures and Tables

**Figure 1 sensors-23-09016-f001:**
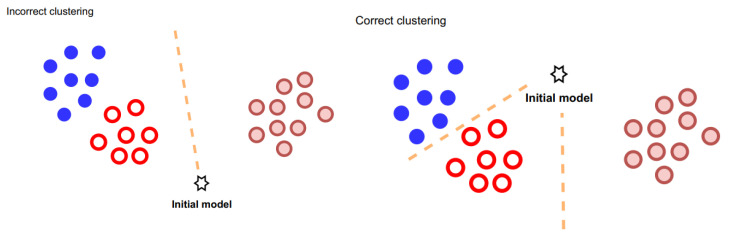
Effect of initial model location on clustering.

**Figure 2 sensors-23-09016-f002:**
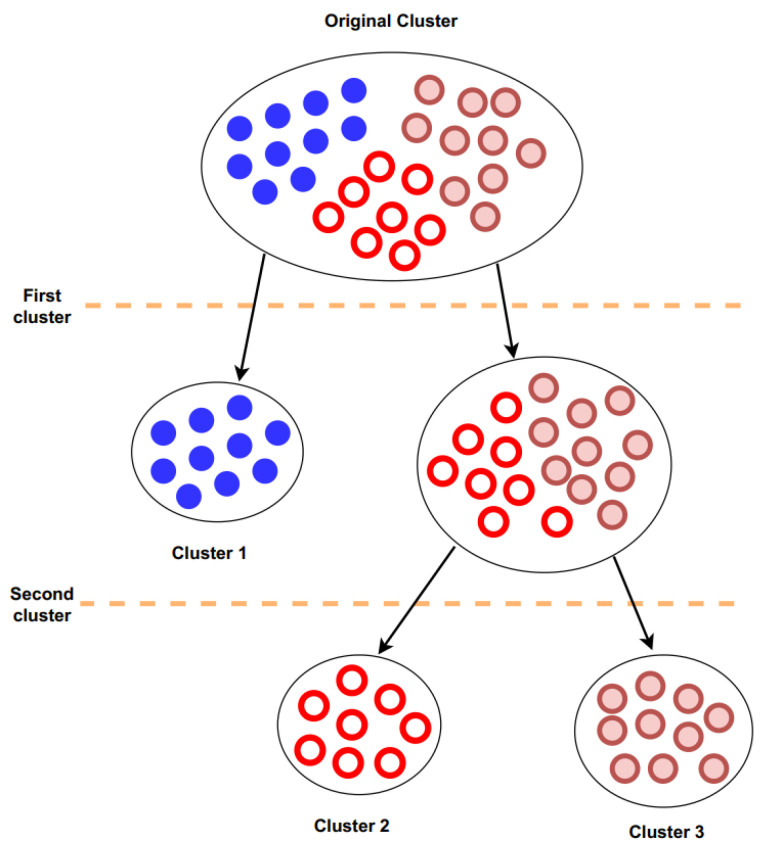
The process of federated adaptive cluster.

**Figure 3 sensors-23-09016-f003:**
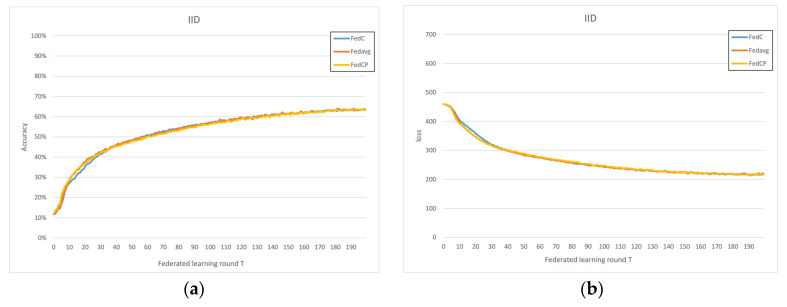
(**a**) Homogeneous IoT data accuracy; (**b**) homogeneous IoT data loss.

**Figure 4 sensors-23-09016-f004:**
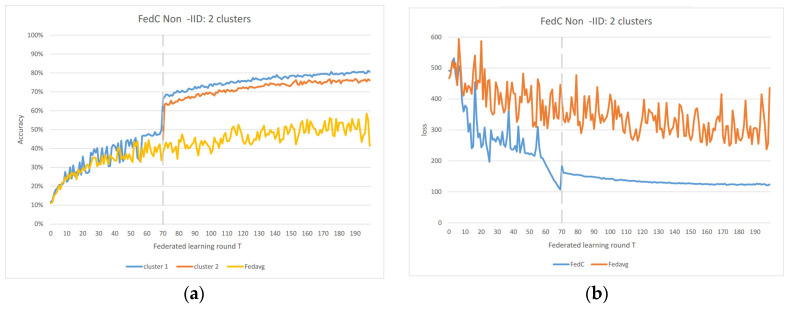
(**a**) FedAC 2 clusters IoT data accuracy; (**b**) FedAC 2 clusters IoT data loss.

**Figure 5 sensors-23-09016-f005:**
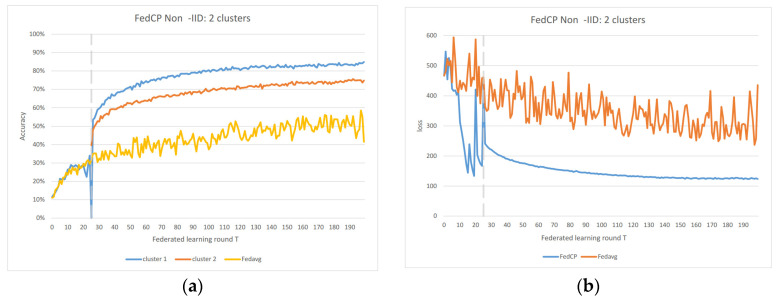
(**a**) FedACP 2 clusters IoT data accuracy; (**b**) FedACP 2 clusters IoT data loss.

**Figure 6 sensors-23-09016-f006:**
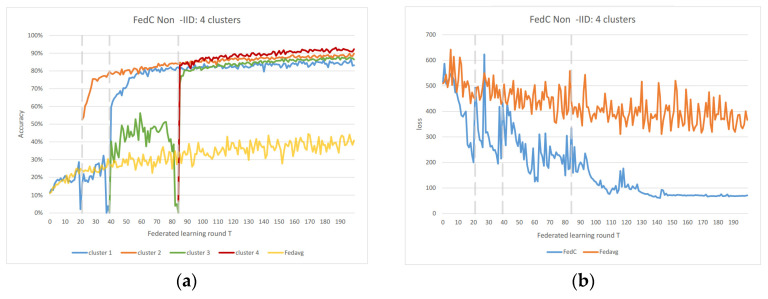
(**a**) FedAC 4 clusters IoT data accuracy; (**b**) FedAC 4 clusters IoT data loss.

**Figure 7 sensors-23-09016-f007:**
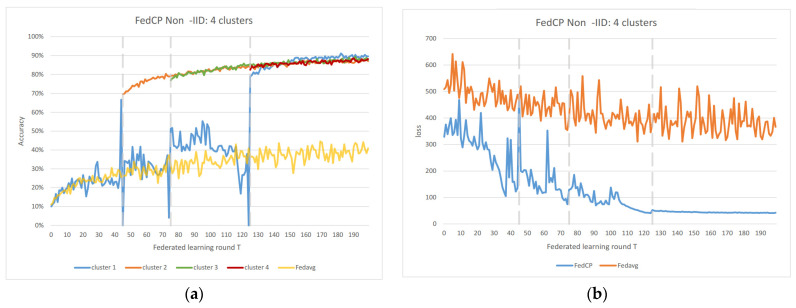
(**a**) FedACP 4 clusters IoT data accuracy; (**b**) FedACP 4 clusters IoT data loss.

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
