# Peer review of "Personalized Federated Learning Algorithm with Adaptive Clustering for Non-IID IoT Data Incorporating Multi-Task Learning and Neural Network Model Characteristics"

_sensors, 2023, doi:10.3390/s23229016_

Round 1

Reviewer 1 Report

Comments and Suggestions for Authors

This paper tackles the intricate issues surrounding federated learning in the context of IoT, a domain where data privacy and heterogeneity are paramount concerns. It introduces a cutting-edge approach that combines personalized joint learning and automatic clustering, effectively addressing the challenges posed by non-IID (Non-Independently and Identically Distributed) IoT data. By doing so, it overcomes the limitations of conventional clustering methods and eliminates the need for rigid cluster specifications. The experimental results highlight the exceptional performance of this approach, particularly in scenarios with specific client distributions. The paper needs some improvements in its current form like:

-the abstract is very long

-the problem formulation in the introduction is not explained. Show some needs of your solution based on the current state of art. See latests solution from the last 2 years and discuss their input in the federated learning area: a hybridization of distributed policy and heuristic augmentation for improving federated learning approach; fuzzy consensus with federated learning method in systems

-explain the novelty in more detials 

-background should be extended to latest solutions.

-your solution needs some better visualization

-add some time/computational complexities analysis 

-experimental section needs more statistical analysis of results

-comparison with state of art is missing

-did you test other types of network?

-how did you select the parameters? 

-security aspects should be discussed 

-is it possible to use your solution in decentralized and centralized models?

Author Response

First of all, we would like to thank the editor for handling our paper and the reviewers for their hard work and constructive comments. We found the reviews very helpful in further improving the quality of this manuscript. The attachment is a list of item-by-item responses to the review comments.

Reviewer 2 Report

Comments and Suggestions for Authors

In this study, the authors present an innovative federated learning-based clustering algorithm designed to address the challenge of Non-IID data in federated learning. The proposed approach aims to enhance the effectiveness and privacy of machine learning in non-IID IoT data.  The study achieves this by clustering clients with similar data distributions together. According to the study's results, the proposed algorithm matches FedAvg in IID scenarios without significantly increasing system load. In non-IID scenarios, the proposed method significantly boosts accuracy and speeds up convergence after clustering with minimal additional system load.

The paper is well organized, presenting clear organization and insightful graphical representations to demonstrate the results of the FedAC algorithm when compared to other algorithms, namely FedACP and FedAvg, across both IID (Independently and Identically Distributed) and non-IID (Non-Independently and Identically Distributed) scenarios.

The way the cosine similarity is computed could be better exemplified.

Author Response

(The authors gave the same response as above.)

Round 2

Reviewer 1 Report

Comments and Suggestions for Authors

I cannot agree that the paper was correctly improved. Please, rewrite the background in the first two sections to the current state of art, which is the last 2-3 years mainly. I propose to analyze federated learning solutions like: a hybridization of distributed policy and heuristic augmentation for improving federated learning approach; fuzzy consensus with federated learning method in systems. 

Such analysis is important due to showing the validation of your proposal in 2023. 

Author Response

Thank you for your valuable feedback. We have revisited the paper and revised the first two sections as per your suggestion to align them with the current state of the art, with a primary focus on the last 1-2 years. Our analysis now encompasses the examination of contemporary federated learning solutions, specifically the application of ”hybridizations of distributed policy and heuristic augmentation” to enhance the federated learning approach and the integration of ”fuzzy consensus” within federated learning methods in systems. 

Round 3

Reviewer 1 Report

Comments and Suggestions for Authors

The paper is ready for publication. 

Author Response

Thank you for your feedback. We are pleased to hear that the paper is ready for publication. We will proceed with the necessary steps to prepare it for submission.